# Early Season Growth Responses of Resistant and Susceptible Cotton Genotypes to Reniform Nematode and Soil Potassium Application

Bhupinder Singh [1,*,†] , Daryl R. Chastain [2], Salliana R. Stetina [3] , Emile S. Gardiner [4] and John L. Snider [5]

1   Delta Research and Extension Center, Mississippi State University, Stoneville, MS 38776, USA
2   USDA Agricultural Research Service, Sustainable Water Management Research Unit, Stoneville, MS 38776, USA
3   USDA Agricultural Research Service, Crop Genetics Research Unit, Stoneville, MS 38776, USA
4   USDA Forest Service, Southern Research Station, Center for Bottomland Hardwoods Research, Stoneville, MS 38776, USA
5   Department of Crop and Soil Sciences, University of Georgia, 115 Coastal Way, Tifton, GA 31794, USA
*   Correspondence: bhupinder.singh@ag.tamu.edu; Tel.: +1-662-518-0559
†   The current address of the senior author is Texas A & M AgriLife Research, 11708 Highway 70 South, Vernon, TX 76385, USA.

**Abstract:** A greenhouse study was conducted to investigate the roles that host plant resistance and soil potassium (K) levels play in affecting *Rotylenchulus reniformis* Linford and Oliveira (Tylenchida: Hoplolaimidae) (RN) populations and early season cotton (*Gossypium hirsutum* L.) growth. Two upland, RN-resistant cotton lines (*G. barbadense* introgressions: 08SS110-NE06.OP and 08SS100), a genetic standard (Deltapine 16) and a commercially available susceptible cultivar (PHY 490 W3FE) were evaluated at four different levels of K [100% of recommended rate, 150% of recommended, 50% of recommended, and a base level] from seeding until harvesting, 60 days after sowing (DAS). Quadratic functions ($r^2 = 0.82$ to 0.95) best described the early season growth response of cotton genotypes to soil K. The base K level was associated with the lowest values for most morphological variables, including plant height (PH), mainstem nodes (MSN), leaf area, and dry weight at 30 DAS and 60 DAS. However, soil K did not affect RN population counts (RC). Additionally, soil K did not influence the rate of change in growth variables among genotypes. The resistant genotype 08SS110-NE06.OP showed greater growth in terms of time to first true leaf, PH, MSN, and above-ground dry weights compared to the commercially available susceptible genotype. No interaction between K and RN or genotype and RN was found in early season cotton growth. However, RC in pots of resistant genotypes was less than in pots of susceptible genotypes. Our research on the early season growth response to soil K by novel, RN-resistant genotypes and susceptible genotypes contributes to the development of improved RN resistance and fertilization management in cotton.

**Keywords:** cotton; resistance; susceptible; potassium; reniform nematode; US Mid-South

## 1. Introduction

Potassium (K) is essential to cotton (*Gossypium hirsutum* L.) production and fiber quality because it plays a direct role in maintaining physiological and metabolic processes critical during all life stages of the plant. It is involved in enzyme activation, water relations, stomatal and non-stomatal limitations to photosynthesis, canopy light interception, internode growth, assimilate translocation, disease resistance [e.g., *Cercospora* (Capnodiales: Mycosphaerellaceae), *Alternaria* (Pleosporales: Pleosporaceae), and *Stemphylium* (Pleosporales: Pleosporaceae) leaf spot], and other nutrient efficiency functions [1–3]. High K uptake occurs during peak bloom and the boll filling period [2,3], and it is closely associated with pollen germination and tube growth, flower opening, seed development (oil percent and seed weight), and fiber quality [4–8]. Research on early maturing cotton

cultivars demonstrates speedy maturity and fruiting in response to K fertilization [5,6]. In contrast, K deficiency in cotton is typically marked by rust or bronze-colored leaves with necrotic margins and premature boll shed [4,9,10]; and, excessive K fertilization can lead to salinity issues, excessive vegetative growth, increased $CO_2$ conductance by mesophyll tissue, increased upper-canopy light interception, and delayed maturity [1,11].

*Rotylenchulus reniformis* Linford and Oliveira (Tylenchida: Hoplolaimidae), reniform nematode (RN) feeds on the roots of cotton seedlings, thereby reducing root surface area and subsequent absorption of relatively immobile soil nutrients such as K [12]. Modifying fertilizer management has been used to compensate for RN damage to cotton growth and development. Interestingly, symptoms of RN damage on cotton mimic K deficiency [12], so earlier studies have evaluated RN damage to cotton across a range of K fertilization [13–15]. However, past research has found a single effect or interacting effects of soil K and RN on cotton growth and development. For instance, Mitchell and Gazaway [13] observed that K deficiency and RN infestation caused cotton stunting, but these effects did not interact. Additionally, RN did not interact with soil K to influence K uptake by cotton. So, K-related deficiency in cotton was not associated with RN. Similarly, Kularathna et al. [15] found that RN reproduction was not affected by K fertilization, nor was there an interaction effect of RN and K on height and biomass of cotton plants. In contrast, Pettigrew et al. [14] concluded that robust root growth from K supplementation supported 12% larger post-harvest RN populations. Genotypic differences for supporting RN populations have also been recognized in cotton. No differences for RN population density were observed among nine genotypes at the early bloom stage (average 2767 RN $kg^{-1}$ soil); however, RN population density differed among genotypes at harvest (9608 to 13,188 RN $kg^{-1}$ soil).

Inconsistent control of RN damage through cultural and chemical strategies has prompted the need to develop cotton cultivars with high RN resistance [16–19]. However, variation in reproduction and pathogenicity of geographic isolates of RN has slowed the deployment of available RN resistance in cotton germplasm. After decades of breeding to improve RN resistance, PhytoGen® released two upland cotton varieties, PHY 332 W3FE and PHY 443 W3FE, in 2021 that show genetic resistance to RN. Similarly, Deltapine® released cultivars DP 2141NR B3XF and DP 2143NR B3XF because of their resistance to RN. The screening efforts used to determine resistance to RN in cotton germplasm lines were mostly based on RN reproduction [20–23]. A large number of greenhouse and field studies have extensively assessed agronomic characteristics to evaluate cotton germplasm under management practices such as soil fertility, soil moisture, air temperature, carbon dioxide, UV-B, nematode damage, herbicide drift, tillage, cover cropping, and row spacing [5,24–35]. Often these studies assessed shoot and root characteristics such as plant height, mainstem node number, leaf thickness and area, dry biomass, and yield. Some work has also assessed how fertilizer modification influences crop resistance to nematode (*Meloidogyne incognita*) infestation [36,37]. However, we do not know if a modification in K fertilization affects cotton resistance to RN as indicated by plant growth and morphology as well as RN fecundity. The evaluation of plant growth traits along with nematode reproduction for a range of K fertilization could yield implications for adaptability and resistance of current and developing cotton germplasm in regions of high nematode occurrence.

The objective of the present study was to evaluate early season cotton growth and RN population development on RN resistant and susceptible genotypes grown in a range of soil K levels. The purpose was to quantify singular or interacting effects of genotype and soil K on RN resistance. We hypothesized that K supplementation would increase early season cotton vigor and decrease RN population size, subsequently decreasing RN damage to cotton. We also hypothesized that resistant genotypes would exhibit greater suppression of RN populations than a commercially available susceptible cultivar, but the two factors, soil K and genotype, would interact to influence the resistance response.

## 2. Materials and Methods

### 2.1. Plant Growth Conditions

The study consisted of four upland cotton lines (*G. barbadense* accession GB 713 introgressions 08SS110-NE06.OP and 08SS100), genetic standard (Deltapine 16; Delta and Pine Land Company, Scott, MS, USA), and commercial cultivar (PHY 490 W3FE; Dow AgroSciences LLC, Indianapolis, IN, USA), which were evaluated at four levels of K with inoculation and without inoculation of RN in a greenhouse. The greenhouse environment was controlled from sowing till harvesting (60 days after sowing) with an electronic control system and environmental variables were monitored with a LI-1400 data logger (Li-Cor, Lincoln, NE, USA), Vaisala HMP50 relative humidity and temperature sensor (Campbell Scientific, Logan, UT, USA), and Li-Cor LI-190 quantum sensor. Air temperature averaged $30 \pm 5$ °C, relative humidity averaged 60%, and photosynthetically active radiation averaged 1500 µmol photons $m^{-2}$ $s^{-1}$ for an 8 h diurnal period. At the time of sowing, four seeds of each genotype were planted in 3 kg volume PVC plastic pots (18 cm diameter $\times$ 15.5 cm length) filled with a steam-treated (70 °C for 8 h) soil media composed of two parts sand and one part Bosket very fine sandy loam soil. Soil media was tested and confirmed that no reniform nematodes present before sowing. Before sowing, a nutrient analysis was conducted on the soil medium (Southern Soils Lab, Yazoo City, MS, USA), which revealed it to be very low in K (29 mg $kg^{-1}$ suggesting a recommended fertilization rate for K of 201 kg $ha^{-1}$ or 90 mg $kg^{-1}$ soil for purposes of this experiment). Muriate of potash (60% $K_2O$) was used for K fertilization. At the time of sowing, each replicate was applied an appropriate concentration of muriate of potash to render four levels of K [100, 150, and 50% of the recommended rate, and base level (0% $K_2O$)]. It was applied at 75, 150, and 225 mg $K_2O$ $kg^{-1}$ soil to develop 50, 100, and 150% levels of recommended rate of K, respectively. Deficiencies of phosphorus (P), nitrogen (N), magnesium (Mg), and sulfur (S) were also reported in the soil medium. The P deficiency was eliminated by amending the soil medium to a recommended level of 20 mg P $kg^{-1}$ soil by applying triple superphosphate at 43.5 mg $kg^{-1}$. Similarly, urea was applied at 130 mg $kg^{-1}$ to meet recommended rates of 60 mg N $kg^{-1}$ soil. Mg and S deficiencies were alleviated by applying magnesium sulfate (15% Mg and 20% S) at a recommended rate of 66.7 mg $kg^{-1}$. Each fertilizer was ground to powder, weighed to the appropriate dose per pot, then dissolved in 100 mL of deionized water. Fertilizer solution was applied over the soil surface in each pot when sowing. Soil moisture was monitored every 15 min with four replicates of moisture sensors (METER Teros 21 sensors; Meter Group Inc., Pullman, WA, USA) inserted 10 cm deep in the soil medium of pots assigned the 100% K treatment. Pots were irrigated using automated drip irrigation at 1.9 L $h^{-1}$ as needed to maintain field capacity (30 cbar). Plants were thinned after one week from sowing to one seedling per pot once uniform emergence was obtained.

An isolate of RN was collected at Stoneville, MS, USA, and maintained in a greenhouse on tomato (*Solanum lycopersicon* 'Rutgers') for use in this study. After seedlings were thinned to 1 per pot, i.e., one week after sowing, about 5000 RN (mixed vermiform life stages) suspended in 1 mL of water were pipetted into a 5 cm deep hole adjacent to the seedling in pots assigned to the RN treatment.

### 2.2. Morphological Measurements

Data were collected to quantify time to 50% emergence and time to first true leaf (FTL). Seedling emergence rate (SER) was calculated from the inverse of time to 50% emergence as described by Reddy et al., 2017. At 30 DAS, we measured non-destructive growth traits, including plant height from soil to mainstem apex (PH), mainstem nodes (MSN), and leaf thickness (TH). At the final harvest (60 DAS), we measured PH, MSN, TH, number of reproductive structures (FN), and leaf area per plant (LA) (Li-3100, Li-COR Inc., Lincoln, NE, USA). Harvested leaf, stem, and reproductive structure tissues were dried in a forced-air dryer oven at 80 °C for 48 h before recording dry weights [leaf dry weight (LDW), stem dry weight (SDW), and reproductive structure dry weight (FDW)].

Roots were gently separated from potting medium, washed, and measured for taproot length (TRL). They were then placed in a forced-air dryer oven at 80 °C for 48 h before measuring dry weight (RDW). Total dry weight (TDW) for each plant was calculated by summing LDW, SDW, FDW, and RDW.

### 2.3. Reniform Nematode Population Measurements

A 200 g sample of soil was collected from each pot and processed for RN population analysis using standard elutriation and sucrose centrifugation protocols [38,39]. Standard elutriation is commonly used to extract reniform nematodes from the soil. First, the soil is suspended in a cone of upward-flowing water. Dense soil particles remained at the bottom of the flowing water suspension, while nematodes and less dense objects floated and poured out of the top of the cone and collected in 325-mesh sieve. The residue captured in the sieve was transferred into a 50 mL centrifuge tube. The tube was spin for 5 min at about 1750 rpm in a centrifuge. The supernatant liquid formed in the tube upon centrifugation was poured off. The sugar solution (453 g cane sugar/liter of water) was then added to the tube to reach a balanced weight and thoroughly mixed with the sediments. The tube was then again spin for 1 min in the centrifuge. The supernatant containing the vermiform RN were poured into 325-mesh sieve and thoroughly rinsed with water to remove any sugar. Vermiform RN were collected in a Petri dish for examination. Vermiform RN was counted using an inverted microscope at 40× magnification. The RN population for each 200 g soil sample was recorded and then converted to an RN population count per kg soil for data analysis.

### 2.4. Data Analysis

The study was established as a 3-factor, factorial experiment arranged in a completely randomized design. The three factors included four levels of K, four levels of genotype, and two levels of RN—treatment combinations that were replicated four times. The experiment was repeated once, and data from both repetitions were combined for analysis. Potassium, genotype, and RN treatments were considered fixed effects, and replicate was considered a random effect. Data were analyzed using a mixed-effects ANOVA model in JMP Pro 12.0 (SAS Institute, Cary, NC, USA). Treatment effect means were separated with Fisher's Protected Least Significant Difference ($\alpha = 0.05$). Growth parameters were fitted to the best regression models against soil potassium to determine the effect of K on cotton growth. Based on highest coefficient of determination ($r^2$), the quadratic model (Equation (1)) best described growth responses to the potassium (Equation (1)).

$$Y = a + bx + cx^2 \tag{1}$$

Coefficients a, b, and c are regression constants, Y is the dependent variable, and x is the % recommended rate of K.

### 3. Results

Growth and RN population responses for each sample period are presented below. For brevity, only significant main effects and interaction effects are presented. Main effects are not presented where significant interactions exist. So, the subsequent sections are organized by genotype, K, and RN effects when response variables were significant in the absence of interaction (Table 1).

**Table 1.** The levels of significance associated with genotype, potassium (K), and reniform nematode (RN) treatments and their interactions effects on early season growth variables of cotton genotypes measured at 30 and 60 days after sowing grown under greenhouse conditions.

| Source of Variation | SER | FTL | TH | PH | MSN | TH | PH | MSN | LA | FN | TRL | RC | FDW | RDW | SDW | LDW | TDW |
|---|---|---|---|---|---|---|---|---|---|---|---|---|---|---|---|---|---|
| | | | | 30 Days | | | | | | | 60 Days | | | | | | |
| Genotype | †NS | † ** | ** | ** | *** | NS | *** | *** | NS | NS | NS | ** | ** | ** | ** | † * | ** |
| K | NS | NS | NS | † *** | *** | NS | *** | *** | *** | *** | *** | NS | *** | *** | *** | *** | *** |
| Genotype and K | NS | NS | ** | NS | NS | NS | NS | NS | NS | NS | NS | NS | NS | NS | NS | NS | NS |
| RN | NS | NS | NS | NS | NS | NS | NS | NS | NS | ** | NS | *** | *** | ** | ** | NS | NS |
| Genotype and RN | NS | NS | NS | NS | NS | NS | NS | NS | NS | NS | NS | ** | NS | NS | NS | NS | NS |
| K and RN | NS | NS | NS | NS | NS | NS | NS | NS | NS | NS | NS | NS | NS | NS | NS | NS | NS |
| Genotype and K and RN | NS | NS | NS | NS | NS | NS | NS | NS | NS | NS | NS | NS | NS | NS | NS | NS | NS |

† The significance levels ***, **, * and NS represent $p \leq 0.001$, $p \leq 0.01$, $p \leq 0.05$ and $p > 0.05$ (NS), respectively. Seed emergence rate (SER), time to first true leaf appearance (FTL), leaf thickness (TH), plant height (PH), mainstem node number (MSN), leaf area per plant, (LA), number of reproductive structures (FN), total root length (TRL), reniform nematode population counts (RC), reproductive structure dry weight (FDW), root dry weight (RDW), stem dry weight (SDW), leaf dry weight (LDW), and total dry weight, (TDW).

### 3.1. Seedling Emergence and First True Leaf
Genotype

During the seedling stage, FTL for PHY 490 W3FE was delayed by at least 9 h as compared to other genotypes ($p < 0.001$) (Table 2).

**Table 2.** Plant height (PH) and mainstem node (MSN) of cotton genotypes measured at 30 days after sowing. Data are means±standard deviation ($n = 64$).

| Genotype | FTL | PH | MSN |
|---|---|---|---|
| | **Hours** | **Cm** | **No. Plant$^{-1}$** |
| 08SS110-NE06.OP | 150 ± 26 [b†] (83–217) * | 25.4 ± 7.0 [a] (11–40) | 5.85 ± 1.2 [a] (3–8) |
| 08SS100 | 147 ± 17 [b] (101–169) | 23.5 ± 6.0 [b] (6–32) | 5.40 ± 1.1 [bc] (3–7) |
| Deltapine 16 | 151 ± 17 [b] (101–193) | 23.6 ± 7.1 [b] (11–40) | 5.66 ± 1.4 [ab] (2–8) |
| PHY 490 W3FE | 160 ± 20 [a] (97–221) | 22.3 ± 6.0 [b] (9–33) | 5.16 ± 1.2 [c] (2–7) |

† Values in a column sharing a lowercase letter are not significantly different ($p > 0.05$) for genotype treatment effect. * Values in a parantheisis describe minumum and maximum values observed for a parameter. Lowercase letters denote statistically significant differences between treatment levels.

### 3.2. 30-Day Measurements
#### 3.2.1. Genotype

PH of 08SS110-NE06.OP was 9.6% greater than the average PH (23.16 cm) for the other three genotypes (Table 2). MSN for 08SS110-NE06.OP (5.8) was greater than for PHY 490 W3FE and 08SS100, but differences were not observed between 08SS110-NE06.OP and Deltapine 16, Deltapine 16 and 08SS100, or 08SS100 and PHY 490 W3FE (Table 2).

#### 3.2.2. K

Base-level K limited PH by 38.2% and MSN by 36.7% relative to the average for higher K levels (Figure 1). In quadratic relations, the PH increased at highest rate of 0.2 cm per percent increase in K, and MSN increased at highest rate of 0.05 nodes per percent increase in K (Figure 1) between 0 and 50% K.

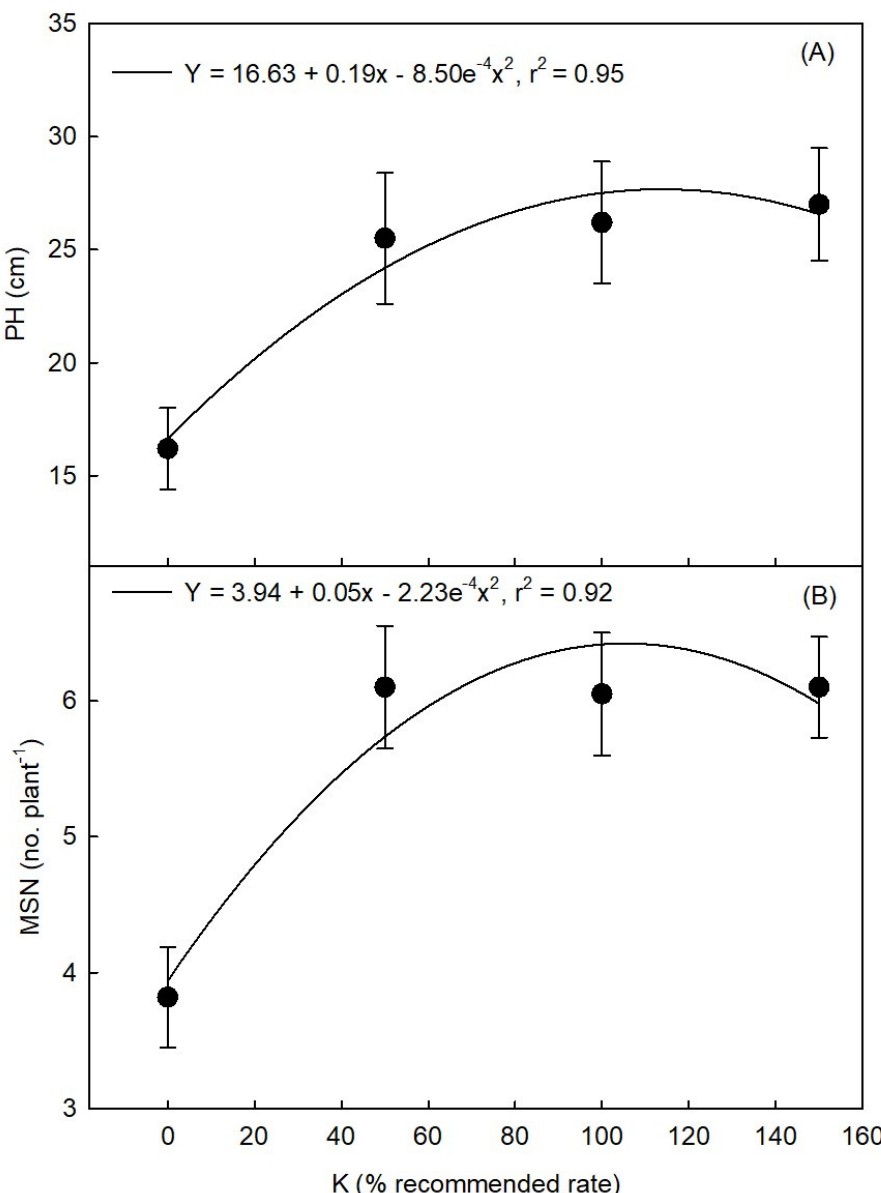

**Figure 1.** K treatment effect on plant height (PH) (**A**) and mainstem nodes (MSN) (**B**) of cotton genotypes at 30 days after sowing. Rates of K application were 150 mg $K_2O$ $kg^{-1}$ soil (recommended rate); 75 mg $K_2O$ $kg^{-1}$ soil (50% of recommended rate); 225 mg $K_2O$ $kg^{-1}$ soil (150% of recommended rate). Data are means ± SE (standard error of the mean) (*n* = 64).

### 3.2.3. Genotype × K Interaction

Unlike other genotypes, 08SS100 had thin leaves (8.8 mm) at K lower than the recommended rate (9.6 mm).

### 3.3. 60-Day Measurements
#### 3.3.1. Genotype

PH ranged from 32.2 cm in PHY 490 W3FE to 38.7 cm in 08SS110-NE06.OP (Table 3). MSN for 08SS110-NE06.OP was 12.1% greater than the average MSN of the other three genotypes (7.14). Resistant genotypes developed heavier RDWs than susceptible genotypes. SDWs and LDWs of resistant genotypes were greater than those of PHY 490 W3FE, but did not different from those of Deltapine 16. FDW did not differ between PHY 490 W3FE and 08SS110-NE06.OP, and averaged 0.23 g. This was lower than FDW for 08SS100 and

Deltapine 16 (0.14 g). PHY 490 W3FE exhibited the lowest TDW among the four genotypes (Table 3).

**Table 3.** Plant height (PH), main stem node number (MSN), root dry weight (RDW), stem dry weight (SDW), fruit dry weight (FDW), leaf dry weight (LDW), total dry weight, (TDW) of cotton genotypes measured at 60 days after sowing. Data are means ± standard deviation (*n* = 64).

| Genotype | PH | MSN | RDW | SDW | FDW | LDW | TDW |
|---|---|---|---|---|---|---|---|
| | **Cm** | **No. Plant⁻¹** | | | **G Plant⁻¹** | | |
| 08SS110-NE06.OP | 38.7 ± 12.5 ᵃ† (14–62) * | 8.13 ± 1.8 ᵃ (4–11) | 1.85 ± 0.8 ᵃ (0.3–2.9) | 3.30 ± 1.8 ᵃ (0.1–7.1) | 0.23 ± 0.1 ᵃᵇ (0–2.1) | 3.35 ± 1.4 ᵃ (0.3–6.1) | 8.50 ± 4.0 ᵃ (0.7–15.3) |
| 08SS100 | 38.1 ± 11.1 ᵇ (12–57) | 7.26 ± 1.6 ᵇ (3–10) | 1.59 ± 0.7 ᵃ (0.1–3.1) | 3.08 ± 1.8 ᵃ (0.3–7.1) | 0.59 ± 0.2 ᵃ (0–2.9) | 3.29 ± 1.6 ᵃ (0–5.6) | 8.58 ± 4.2 ᵃ (0–14.8) |
| Deltapine 16 | 33.2 ± 10.2 ᵇᶜ (13–57) | 7.10 ± 1.6 ᵇ (2–9) | 1.80 ± 1 ᵇ (0.2–5.6) | 2.89 ± 1.8 ᵃᵇ (0.2–8.0) | 0.34 ± 0.1 ᵃ (0–2.8) | 3.11 ± 1.5 ᵃᵇ (0.3–6.16) | 8.13 ± 4.3 ᵃ (0.8–19.5) |
| PHY 490 W3FE | 32.2 ± 10.0 ᶜ (10–57) | 7.06 ± 1.7 ᵇ (3–11) | 1.54 ± 0.7 ᵇ (0.2–2.8) | 2.48 ± 1.4 ᵇ (0.2–5.7) | 0.22 ± 0.1 ᵇ (0–1.7) | 2.87 ± 1.4 ᵇ (0.2–6.9) | 7.14 ± 3.5 ᵇ (0.7–14.8) |

† Values in a column sharing a letter are not significantly different (*p* > 0.05) for genotype treatment effect. * Values in a parantheisis describe minumum and maximum values observed for a parameter. Lowercase letters denote statistically significant differences between treatment levels.

### 3.3.2. K

Quadratic relationships were observed between K and various growth traits, including PH, MSN, LA, FDW, SDW, LDW, RDW, TRL, and TDW (Figures 2–4). Base-level K produced the shortest PH (20.1 cm) which averaged 10 cm shorter than that of plants grown at higher K levels (Figure 2A). Similarly, MSN at base-level K was lower than MSN for other K levels (Figure 2B). Development of leaf area at base-level K was 72% less than at higher K levels (Figure 2C). In the quadratic relations, PH increased at highest rate by 0.39 cm, MSN by 0.06 nodes, and LA by 0.22 cm² per percent increment in K between 0 and 50% K.

Similar trends in response to K were observed for dry weight variables (Figure 3A–C). For instance, TDW was most limited at base-level K (76% lower than other levels), and increased at highest rate by 0.16 g per percent increment in K between 0 and 50% K.

TRL increased between K levels of 0 and 50%, but no change was observed beyond 50% K. The TPL increased at highest rate by 0.16 cm per percent increment in K between 0 and 50% K (Figure 4). RN reproduction was not influenced by K (Table 1).

### 3.4. Reniform Nematode

FN decreased by 42% and FDW decreased 62% under inoculation of RN (Table 4). SDW decreased 14%, while RDW increased 11.6% under inoculation of RN.

**Table 4.** Reniform nematode (RN) treatment effects on number of reproductive structures (FN), root dry weight (RDW), stem dry weight (SDW), and fruit dry weight (FDW) of cotton genotypes at 60 days after sowing. Data are means (*n* = 128).

| RN | FN | FDW | RDW | SDW |
|---|---|---|---|---|
| | **No. Plant⁻¹** | | **G Plant⁻¹** | |
| Without incoulation | 0.65 ± 0.2 ᵃ† (0–3) | 0.52 ± 0.3 ᵃ (0–2.9) | 1.59 ± 0.7 ᵇ (0–0.2) | 3.16 ± 1.9 ᵃ (0–7.9) |
| With inoculation | 0.38 ± 0.1 ᵇ (0–2) | 0.19 ± 0.1 ᵇ (0–2.4) | 1.81 ± 0.8 ᵃ (0–0.1) | 2.72 ± 1.6 ᵇ (0.2–6.5) |

† Values in a column sharing a letter are not significantly different (*p* > 0.05) for the interaction effect. Lowercase letters denote statistically significant differences between treatment levels.

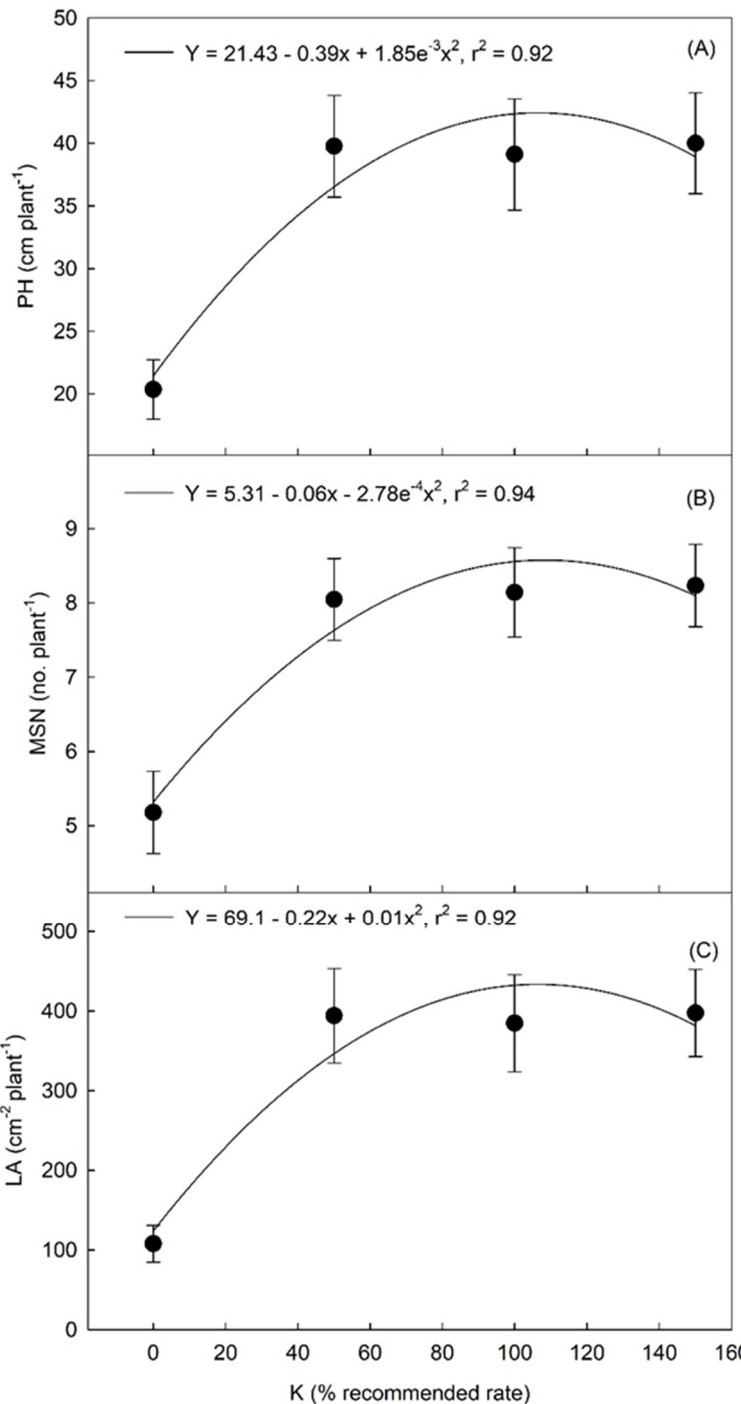

**Figure 2.** K treatment effect on plant height (PH) (**A**), mainstem nodes (MSN) (**B**), and leaf area (LA) (**C**) of cotton genotypes at 60 days after sowing. Rates of K application were 150 mg $K_2O$ $kg^{-1}$ soil (recommended rate); 75 mg $K_2O$ $kg^{-1}$ soil (50% of recommended rate); 225 mg $K_2O$ $kg^{-1}$ soil (150% of recommended rate). Data are means $\pm$ SE (standard error of the mean) ($n = 64$).

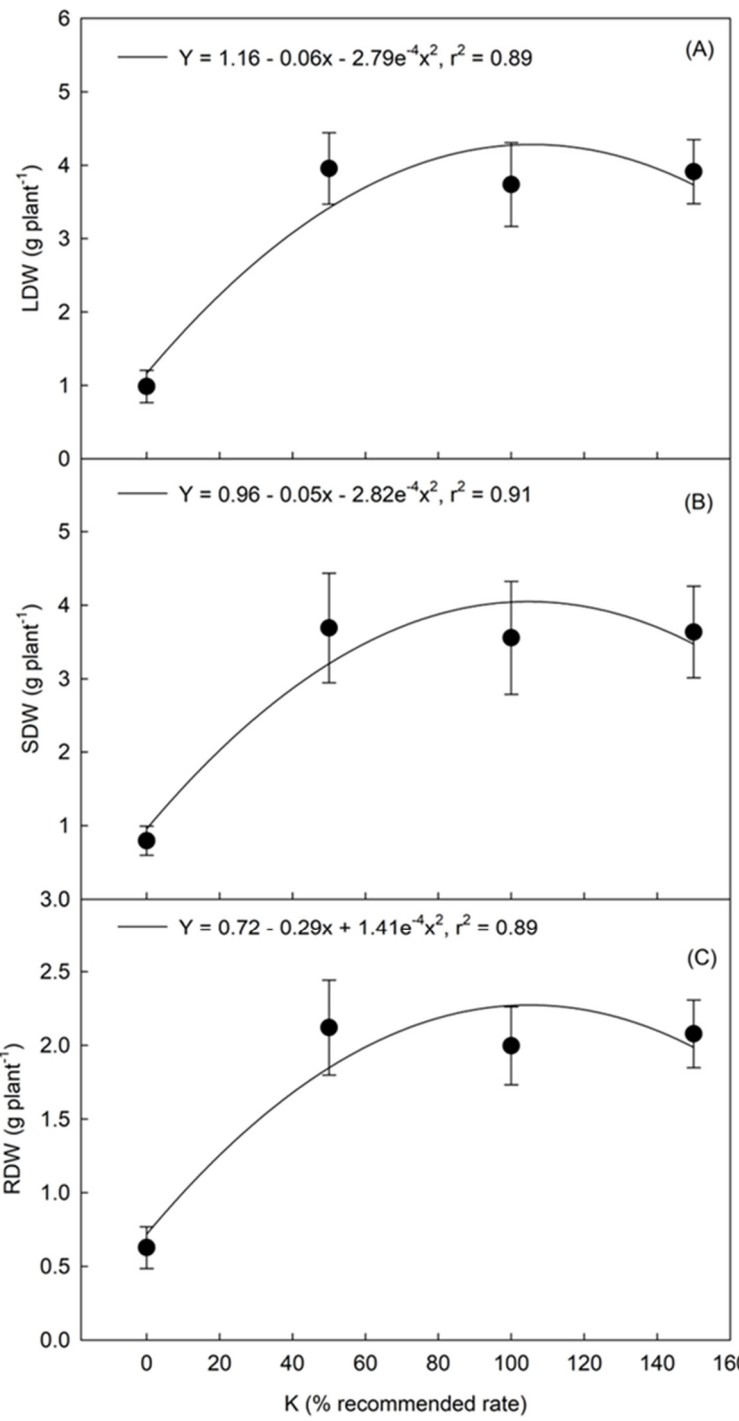

**Figure 3.** K treatment effect on leaf (LDW) (**A**), stem (SDW) (**B**), and root (RDW) (**C**) dry weights of cotton genotypes at 60 days after sowing. Rates of K application were 150 mg $K_2O$ kg$^{-1}$ soil (recommended rate); 75 mg $K_2O$ kg$^{-1}$ soil (50% of recommended rate); 225 mg $K_2O$ kg$^{-1}$ soil (150% of recommended rate). Data are means $\pm$ SE (standard error of the mean) ($n$ = 64).

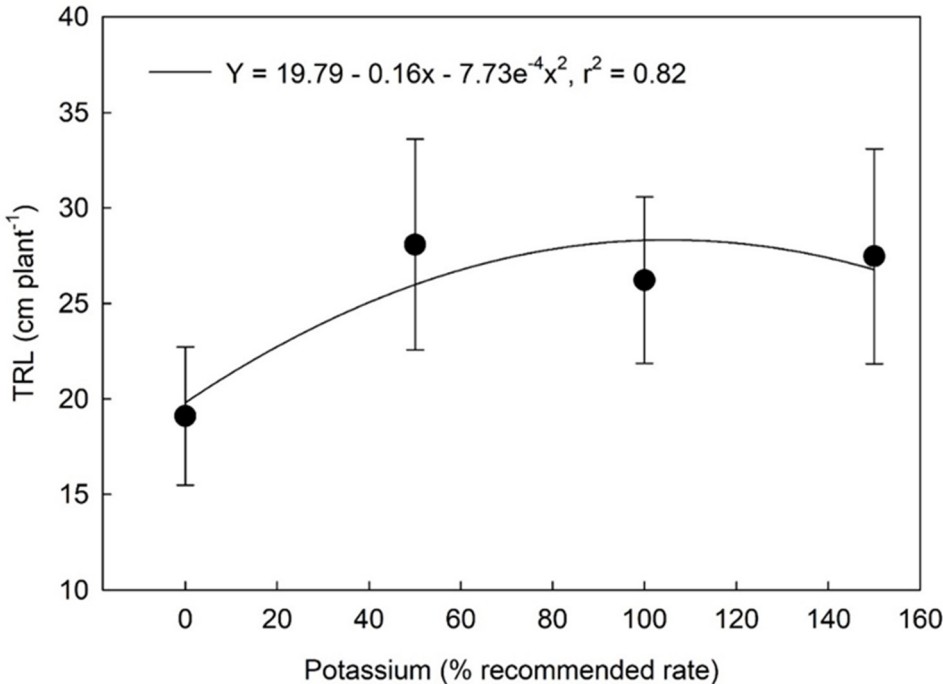

**Figure 4.** K treatment effect on taproot length (TRL) of cotton genotypes at 60 days after sowing. Rates of K application were 150 mg $K_2O$ $kg^{-1}$ soil (recommended rate); 75 mg $K_2O$ $kg^{-1}$ soil (50% of recommended rate); 225 mg $K_2O$ $kg^{-1}$ soil (150% of recommended rate). Data are means ± SE (standard error of the mean) ($n = 64$).

### 3.5. Interaction

A genotype by RN interaction was only observed for RN population count. For pots that received RN inoculation, those maintaining the resistant genotype 08SS110-NE06.OP supported an 81% lower RN population count than those maintaining the other genotypes (Table 5).

**Table 5.** Interaction effect of genotype (G) and reniform nematode (RN) treatments on reniform nematode population count (RC) in pots grown with cotton genotypes at 60 days after sowing. Data are means±standard deviation ($n = 8$).

| Genotype | RC Nematodes $kg^{-1}$ | |
|---|---|---|
| | **With Inoculation** | **Without Inoculation** |
| Deltapine 16 | 9793 ± 1199 [a†] | 866 ± 297 [b] |
| PHY 490 W3FE | 8891 ± 952 [a] | 1576 ± 73 [b] |
| 08SS100 | 5787 ± 748 [a] | 360 ± 62 [b] |
| 08SS110-NE06.OP | 1744 ± 439 [b] | 974 ± 236 [b] |

† Values in columns and rows sharing a letter are not significantly different ($p > 0.05$). Lowercase letters denote statistically significant differences between treatment levels.

## 4. Discussion

We evaluated early season growth of RN resistant and RN susceptible cotton genotypes raised with and without inoculation of RN. To our knowledge, this work is the first to investigate the impact of K on growth and RN population suppression by resistant genotypes 08SS110-NE06.OP and 08SS110. Previous research on growth and development of cotton and other major crops has largely focused on assessing responses to abiotic stresses, including cold, heat, drought, UV-B, and carbon dioxide [31,33,35,40]. Cotton response to biotic stress, such as RN, has received less attention [25,41,42]. The current study confirmed the effects of RN and K treatments on plant height at 30 and 60 DAP, which

agrees with Mitchell and Gazaway [13] and Kularathna et al. [15]. These studies found cotton stunting associated with K deficiency or RN infection. Mitchell and Gazaway [13] observed aboveground dry weight to respond positively to K fertilization. However, Kularathna et al. [15] did not find an impact of K fertilization on the shoot or root biomass. One reason could be that the soil used by Kularathna et al. [15] held a higher K (44 mg kg$^{-1}$) control level than the soil in the present study. Plants in the current study showed an increase in shoot biomass between 0 and 50% of the recommended rate of K. The higher root biomass and taproot length we observed for plants grown in K supplemented soil agrees with the improved root vigor Pettigrew et al. [14] observed in response to K. Thus, K fertilization appeared essential to optimize growth and development of the cotton genotypes researched in the present study.

Root processing conducted in this study was performed meticulously to avoid introducing errors that could obscure RN damage [13]. RN affected root biomass differently than shoot biomass as root biomass was relatively higher than shoot biomass under RN inoculation. The resulting higher root to shoot ratio has been recognized as a characteristic symptom of RN damage in cotton [43]. Yet, an increased root length could reduce the parasitic load of RN and the resulting yield loss [44]. Therefore, a management practice, such as fertilization, that accelerates early taproot growth could be an effective approach to limiting RN damage on cotton.

Unlike cotton growth traits, RC did not respond to K fertilization—this finding is supported by those of Mitchell and Gazaway [13] and Kularathna et al. [15]. In contrast to the response observed for the K treatment, earlier work on RN showed either positive or negative chemotactic population responses to inorganic compounds such as urea, ammonium salts, and ammonium nitrate [45,46]. Pettigrew et al. [14] reported a high RN population under K fertilization, but that observation contradicts this work and other published studies [13,15]. The unrestricted vertical or/and horizontal movement of RN under field conditions [18] could explain why the results of Pettigrew et al. [14] contrast with our results.

K and RN did not interact to influence the cotton growth variables we measured. This finding confirmed the conclusion by Mitchell and Gazaway [13] and Kularathna et al. [15], who concluded that a high level of K would not reduce RN damage on early season growth of cotton. Evaluation of cotton growth in response to K and RN in the same experiment provided us the opportunity to assess some shared symptoms of RN damage and K deficiency. As observed in this study, stunted growth with reduced MSN, plant biomass, and flowering and fruit set are common symptoms shared between K deficiency and RN damage [5,12,25,27,41,47–49]. Similar to K effects in the present study, growth traits such as plant height, mainstem nodes, above and below-ground biomass, and TPL were successfully used to quantify abiotic and biotic stress tolerance in cotton and other agronomic crops during early growth stages [31,33,35,40].

RN resistant cotton genotype 08SS110-NE06.OP suppressed RC in the current study, which means that using these genotypes can offer a level of RN control. RN population was extracted using standard elutriation and sucrose centrifugation techniques. These techniques involve several steps such as wet sieving of soil samples, soil suspension in water, agitating the suspension, and counting nematodes in the slides. Although intense care was taken to clean each equipment to eliminate cross contamination between samples thoroughly, it is impossible to completely avoid cross contamination between samples during the process. Therefore, some contamination between samples may have led to reniform nematode numbers observed in soil samples without inoculation. Additionally, RC observed in uninoculated pots were lower than economic threshold for reniform nematode infestation on cotton in light textured soils (i.e., 2000 RN per kg soil). Thus, the cross contamination that occurred was not at the level necessary to induce RN effects on measured parameters. A recent study by Galbieri et al. [44] reported a linear relationship between RN infestation and cotton yields among 12 cotton genotypes. The relationship revealed the highest yields were associated with genotypes that supported the lowest RC

and were characterized as most tolerant. Thus, high resistance to RN will likely contribute to high and stable yields across RN environments. The interaction effect of K by genotype was not observed for RC. This suggests the variation in RC we observed was solely due to plant resistance and not K supplementation, and it also indicates K supplementation may not modify the inherent sensitivity, tolerance, or resistance of host plants to RN.

Previously, RN resistant lines have been shown to limit RN fecundity [21,22,50], but their performance in inhibiting RN reproduction varied across the diverse pathogenicity of RN isolates [23]. This variable response to pathogenicity has hindered the deployment of available RN resistance in cotton germplasm. In fact, it was very recent (2021) when PhytoGen® cottonseed and Deltapine® released varieties that demonstrate resistance to RN. Future studies should address the performance of resistant genotypes used in this study across the diverse pathogenicity among RN isolates. Our observations of high RC and reduced growth under the RN inoculation confirm that the commercial check used in this study was susceptible to RN. Unlike RC, a genotype by RN interaction was not observed for the various growth variables. This would suggest that the variables of growth we studied may not be strong indices of tolerance during early season growth. Thus, the supposition is supported by a related study that showed comparable early season growth and development among susceptible checks and resistant lines (i.e., sowing to 7 weeks after sowing) under RN infested field conditions [51].

As with the current study, RN and genotype did not interact to influence plant growth and development when resistant cotton lines were tested across a range of N and RN [52]. Nor did we observe a K by genotype interaction, which indicates the early growth response of resistant genotypes to K was similar to that of commercial cultivars. However, earlier studies recognized the breadth of genotypic variation in uptake of soil available K [53,54]. Performance of the novel resistant lines relative to commercial checks under recommended K levels in this study suggests their appropriateness to upland cotton production systems where RN are problematic. However, additional research will be needed to characterize the agronomic performance and resistance mechanisms of the resistant lines across varied management practices and growth stages before finalizing their deployment.

## 5. Conclusions

We observed a significant K fertilization and RN impact on the early season growth of four cotton genotypes examined in this study. However, the genotype did not interact with K and all four genotypes showed a quadratic growth response to K fertilization. This response was such that early season cotton growth leveled off as K rose above 50% of the recommended rate. The growth response functions we present here will improve simulating early season crop growth in dynamic cropping system simulation models. A genotype by RN interaction revealed that resistant genotypes possess a relatively high level of resistance to RN, but the growth variables selected did not reveal the mode of resistance.

**Author Contributions:** Conceptualization, B.S. and S.R.S.; methodology, B.S. and S.R.S.; investigation, B.S., S.R.S. and D.R.C.; writing—original draft, B.S.; writing—review and editing, B.S., D.R.C., S.R.S., B.S. and E.S.G.; supervision, D.R.C.; data curation and visualization, E.S.G.; validation, J.L.S.; formatting and submission, B.S. All authors have read and agreed to the published version of the manuscript.

**Funding:** This research was supported in part by [the U.S. Department of Agriculture (USDA), Agricultural Research Service], under agreement number [58-6066-6-045].

**Data Availability Statement:** Data supported the findings of the study could be acquired from the corresponding author by reasonable request.

**Acknowledgments:** We are very thankful to April Reynolds and graduate students at the Delta Research and Experimentation Stoneville, Mississippi, for technical assistance.

**Conflicts of Interest:** The authors declare no conflict of interest.

**Disclaimer:** The findings and conclusions in this publication are those of the authors and should not be construed to represent any official USDA or U.S. Government determination or policy. Mention of trade names or commercial products in this publication is solely for the purpose of providing specific information and does not imply recommendation or endorsement by the USDA. USDA is an equal opportunity provider and employer. We are very thankful to April Reynolds and graduate students at Delta Research and Experimentation Stoneville, Mississippi for technical assistance.

## Abbreviations

| | |
|---|---|
| FTL | time to first true leaf |
| FDW | weight of reproductive structures |
| FN | number of reproductive structures |
| K | potassium |
| LA | leaf area per plant |
| LDW | dry leaf weight |
| MSN | mainstem nodes, N, nitrogen |
| NO | without inoculation of reniform nematode |
| PH | plant height |
| RN | reniform nematode |
| SER | seedling emergence rate |
| TH | leaf thickness |
| RC | reniform nematode population counts |
| RDW | root dry weight |
| SDW | stem dry weight |
| TRL | taproot length |
| TDW | total dry weight |

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
