# Peer review of "Early Season Growth Responses of Resistant and Susceptible Cotton Genotypes to Reniform Nematode and Soil Potassium Application"

_agronomy, doi:10.3390/agronomy12112895_

Round 1
Reviewer 1 Report
The article draws attention to the important problem: impact of resistance cotton genotypes and potassium (K) fertilization to Reform nematode in early season growing cotton.
My comments and suggestion are indicated below:
Line 167: The detail description of sucrose centrifugation protocols is needed
Line 183: It is necessary to explain why the quadratic equation was chosen to approximate the dependence
Line 198: The order of column names in the table and the order of abbreviations in the table legend must match
Line 196: Title Table 1. Analysis of variance does not match its content. The table contains only tests of influence significance, but not the analysis of variance
Tables 2, 3 and 4 can be combined into one with descriptive statistics of the corresponding variables, which in addition to the mean values characterize their variation
Author Response
Below are my answers to the comments of reviewers’:
Reviewer 1:
The article draws attention to the important problem: impact of resistance cotton genotypes and potassium (K) fertilization to Reform nematode in early season growing cotton.
My comments and suggestion are indicated below:
Line 167: The detail description of sucrose centrifugation protocols is needed’
The details of standard elutriation and sucrose centrifugation methods were added.
Line 183: It is necessary to explain why the quadratic equation was chosen to approximate the dependence
The following changes were made to address this comment:
“Growth parameters were fitted to the best regression models against soil potassium to determine the effect of K on growth. Based on highest coefficient of determination (r2), the quadratic model (Eq. 1) best described growth responses to the potassium (Eq. 1)”
Line 198: The order of column names in the table and the order of abbreviations in the table legend must match
Changes were made as suggested
Line 196: Title Table 1. Analysis of variance does not match its content. The table contains only tests of influence significance, but not the analysis of variance
The Title of the table was changed to match the table contents.
Tables 2, 3 and 4 can be combined into one with descriptive statistics of the corresponding variables, which in addition to the mean values characterize their variation
Table 2 and 3 were combined as they both include 30-day measurements. Table 4 belonged to 60-day measurements and therefore kept separate. Also, combining Table 4 to Table 2 & 3 would make the final table challenging to fit in window width with visible font size.
The standard deviations, minimum and maximum values of the corresponding variables across the treatment factor were added with mean values in the table to characterize the variations.
Reviewer 2 Report
The study is quite interesting and well-designed, although the results are not unexpected. The statistical analyses are quite accurate. The results and discussion sections are enough plain and clear. In my opinion, the MS doesn't require any further revision and is ready to publish in this form.
Author Response
Thanks for reviewing the manuscript.
Reviewer 3 Report
This manuscript evaluates the interaction of K fertilization and reniform nematode in resistant and susceptible genotypes. As a greenhouse trial, it does not provide meaningful yield data, however the growth parameters measured at 30 and 60 days after sowing are useful in interpreting growth responses. The lack of interaction between K fertilization and plant growth responses to K is particularly useful, as it highlights the need for specific nematode control practices.
Specific editorial comments include:
Line 59. Replace "like" with "such as".
Lines 72-75. Awkward sentence structure. Revise to read "No differences for RN population density were observed among nine genotypes at the early bloom stage (average 2767 RN kg-1 soil); however, RN population density differed among genotypes at harvest (9608 to 13188 RN kg-1 soil).".
Lines 119. Please indicate the diameter and height (soil depth) of the plastic pots. This will be important in interpreting the tap root length data.
Lines 141-147. Please indicate the number of Days After Sowing when plant thinning and RN inoculation occurred.
Line 221. The statement that "PH increased by 0.2 cm, and MSN increased 0.05 nodes for each percent increase in K" implies a linear relationship, but the data clearly show a quadratic effect. Virtually all of the increase in these (and other parameters graphed in Figs. 1-4) occurred between the 0 and 50 % K rates. The data appear to indicate a plateau in growth above the 50 % rate, rather than a continuous response. While a quadratic regression is an appropriate way to analyze the response to K, the narrative of results should avoid implying a continuous response unless the quadratic term is negligible.
Figure 1 A and B. Should the regression equations include the "x" after 0.19 and 0.05, respectively? As written, those equations do not make a lot of sense.
Table 4. Please verify the means separation. The conclusion that a Plant Height of 38.7 cm is significantly different from 38.1, but 38.1 is not different from 33.2, seems difficult to understand.
Lines 290-294. It appears from Table 6 that uninoculated plants developed significant RN populations, which indicates that either the number at plants was NOT zero, or that significant cross-contamination among pots occurred. The lack of evidence for resistance in 08SS110-NE06.OP in these uninoculated pots will need some explanation. The discussion at lines 350-357 appears to blame this on cross contaminations during sample processing, however a few nematodes contaminating equipment between samples does not seem to account for the pretty high counts (~10-50 % of the numbers in inoculated pots) reported from uninoculated pots.
Table 6 also appears to show that the cv 08SS100 does not suppress nematode population increases significantly. Some discussion explaining this finding is warranted.
Table 6. Revise the footnote to read "Values in columns and rows sharing a letter are not significantly different (P > 0.05)."
Line 307. References 40-43 do not appear to relate to cotton. Can those be omitted? Alternatively, explain why they are relevant to cotton.
Lines 357-379. The terms "tolerance" and "resistance" must be used carefully. Tolerance refers to the ability of a cultivar to produce high yields in the presence of the pathogen, whereas resistance refers to suppression of pathogen reproduction by the host. It is not uncommon for a resistant host to suffer significant yield loss (low tolerance) and vice versa. It may be useful to clarify these points for the reader. For example, replace the word "resistance" with "tolerance" in line 376.
Line 441. Correct the spelling of "effects".
Lines 475-476. Do not capitalize key words in titles of journal articles. This also applies to references 45 and 56.
Author Response
This manuscript evaluates the interaction of K fertilization and reniform nematode in resistant and susceptible genotypes. As a greenhouse trial, it does not provide meaningful yield data, however the growth parameters measured at 30 and 60 days after sowing are useful in interpreting growth responses. The lack of interaction between K fertilization and plant growth responses to K is particularly useful, as it highlights the need for specific nematode control practices.
Specific editorial comments include:
Line 59. Replace "like" with "such as".
Replaced
Lines 72-75. Awkward sentence structure. Revise to read "No differences for RN population density were observed among nine genotypes at the early bloom stage (average 2767 RN kg-1 soil); however, RN population density differed among genotypes at harvest (9608 to 13188 RN kg-1 soil).".
Revised accordingly
Lines 119. Please indicate the diameter and height (soil depth) of the plastic pots. This will be important in interpreting the tap root length data.
Provided. It was 18 cm diameter x 15.5 cm length.
Lines 141-147. Please indicate the number of Days After Sowing when plant thinning and RN inoculation occurred.
Provided. It was one week after sowing.
Line 221. The statement that "PH increased by 0.2 cm, and MSN increased 0.05 nodes for each percent increase in K" implies a linear relationship, but the data clearly show a quadratic effect. Virtually all of the increase in these (and other parameters graphed in Figs. 1-4) occurred between the 0 and 50 % K rates. The data appear to indicate a plateau in growth above the 50 % rate, rather than a continuous response. While a quadratic regression is an appropriate way to analyze the response to K, the narrative of results should avoid implying a continuous response unless the quadratic term is negligible.
The statements were revised to clearly state that highest rate of increase in a quadratic increase were achieved between 0 and 50 % K.
Figure 1 A and B. Should the regression equations include the "x" after 0.19 and 0.05, respectively? As written, those equations do not make a lot of sense.
Revised accordingly.
Table 4. Please verify the means separation. The conclusion that a Plant Height of 38.7 cm is significantly different from 38.1, but 38.1 is not different from 33.2, seems difficult to understand.
We verified the analysis and It’s correct.
Lines 290-294. It appears from Table 6 that uninoculated plants developed significant RN populations, which indicates that either the number at plants was NOT zero, or that significant cross-contamination among pots occurred. The lack of evidence for resistance in 08SS110-NE06.OP in these uninoculated pots will need some explanation. The discussion at lines 350-357 appears to blame this on cross contaminations during sample processing, however a few nematodes contaminating equipment between samples does not seem to account for the pretty high counts (~10-50 % of the numbers in inoculated pots) reported from uninoculated pots.
The following information was added to the discussion:
The economic threshold of reniform nematode infestation on cotton early to mid-season growth is above 2000 RN per kg soil. The RN counts in uninoculated pots were lower than economic threshold. Conversely, RN counts in inoculated pots were higher than economic thresholds. The cross contamination occurred was not significant to impose a risk of RN effects on measured parameters.
Table 6 also appears to show that the cv 08SS100 does not suppress nematode population increases significantly. Some discussion explaining this finding is warranted.
Although the RN counts of 08SS100 were not statistically lower than susceptible cultivars, however, unlike susceptible cultivars, 08SS100 strictly inhibit RN reproduction in a manner that RN count at the time of harvesting was 5787 per kg soil which is slightly higher than rate of inoculation at time of emergence (5000 per kg soil). We further addressed the variation in the ability to suppress RN reproduction among resistant lines by clearly stating in Lines 381-384 with references.
“Previously, RN resistant lines have been shown to limit RN fecundity [21,22,54], but their performance in inhibiting RN reproduction varied across the diverse pathogenicity of RN isolates [23]. This variable response to pathogenicity has hindered the deployment of available RN resistance in cotton germplasm”
Table 6. Revise the footnote to read "Values in columns and rows sharing a letter are not significantly different (P > 0.05)."
Revised
Line 307. References 40-43 do not appear to relate to cotton. Can those be omitted? Alternatively, explain why they are relevant to cotton.
Removed
Lines 357-379. The terms "tolerance" and "resistance" must be used carefully. Tolerance refers to the ability of a cultivar to produce high yields in the presence of the pathogen, whereas resistance refers to suppression of pathogen reproduction by the host. It is not uncommon for a resistant host to suffer significant yield loss (low tolerance) and vice versa. It may be useful to clarify these points for the reader. For example, replace the word "resistance" with "tolerance" in line 376.
Replaced
Line 441. Correct the spelling of "effects".
This citation was updated as asked by another reviewer
Lines 475-476. Do not capitalize key words in titles of journal articles. This also applies to references 45 and 56.
Changes were made.
Reviewer 4 Report
The manuscript titled “Early-season Growth Responses of Resistant and Susceptible Cotton Genotypes to Reniform Nematode and Soil Potassium Application” has been reviewed. There are some observations that required the attention of the authors before the manuscript can be accepted.
When the name of an animal is included in the title, the scientific name should be followed by the order and family placement.
The full scientific name of an insect species that is being studied must be given together with the authority and the order and family placement when first mentioned in the abstract and the main text.
I recommend including the abstract, general description of methods, novelty, treatments, or evaluations, main results expressed with values and statistical significance, and the conclusion of the evaluation or analysis of the experimental results.
Authors need to make a graphical abstract.
The discussion needs more citations
Update all old citations
Author Response
Below are our answers in red fonts to reviewer's comments:
When the name of an animal is included in the title, the scientific name should be followed by the order and family placement.
The authority, order and family following full scientific name of reinform nematode was added in Title as.: Rotylenchulus reniformis Linford and Oliveira (Tylenchida: Hoplolaimidae)
The full scientific name of an insect species that is being studied must be given together with the authority and the order and family placement when first mentioned in the abstract and the main text.
The authority, order and family following full scientific name of reinform nematode was added in abstract and maintext.: Rotylenchulus reniformis Linford and Oliveira (Tylenchida: Hoplolaimidae)
I recommend including the abstract, general description of methods, novelty, treatments, or evaluations, main results expressed with values and statistical significance, and the conclusion of the evaluation or analysis of the experimental results.
The manuscript has already been organized according to the Journal guidelines and contains information relevant to each section.
Authors need to make a graphical abstract.
Graphical abstracts would have been reasonable if we had any interaction effects among the treatments. However, in this study, treatment factors showed no interactions for the growth and development of cotton. Therefore, adding a graphical abstract would not summarize the study appropriately. Also, graphical abstract is not a requirement for this journal.
The discussion needs more citations
New citations were added to the discussion, and old ones were updated by replacing them with recently published papers.
Update all old citations
The new citations were added to the discussion, and old citations were updated by replacing them with recently published papers.
Round 2
Reviewer 4 Report
I think now the manuscript can publish in Agronomy